# PERMA Model of Well-Being Applied to Portuguese Senior Tourists: A Confirmatory Factor Analysis

José Mendes [1,*], Teresa Medeiros [1], Osvaldo Silva [2], Licínio Tomás [2], Luís Silva [3] and Joaquim A. Ferreira [4]

1   School of Social Sciences and Humanities, University of Azores, 9500-321 Ponta Delgada, Portugal; maria.tp.mendeiros@uac.pt
2   Interdisciplinary Centre of Social Sciences—Campus of the University of Azores (CICS.NOVA.UAc), University of Azores, 9500-321 Ponta Delgada, Portugal; osvaldo.dl.silva@uac.pt (O.S.); licinio.mv.tomas@uac.pt (L.T.)
3   Research Network in Biodiversity and Evolutionary Biology (InBIO), Research Center in Biodiversity and Genetic Resources (CIBIO), School of Sciences and Technology, University of Azores, 9500-321 Ponta Delgada, Portugal; luis.fd.silva@uac.pt
4   Center for Research in Neuropsychology and Cognitive and Behavioral Intervention (CINEICC), Faculty of Psychology and Education, University of Coimbra, 3000-115 Coimbra, Portugal; jferreira@fpce.uc.pt
*   Correspondence: jose.cs.mendes@uac.pt

**Abstract:** Tourism has been affirmed as an activity that promotes health and well-being. The present study aims to present a confirmatory analysis of the PERMA model in a sample of Portuguese senior tourists who visited the island of São Miguel (Azores). After approval of the study by the ethics committee (reference 6/2022), a Sociodemographic Questionnaire, PERMA Profiler, and Life Satisfaction Scale—SWLS were applied to 1083 senior tourists ($\geq$55 years) of various nationalities. To evaluate the PERMA model for senior tourism in Portugal, a total of 434 senior tourists of Portuguese nationality were extracted from the sample. The results revealed that most of the participants attributed scores above the midpoint in the five dimensions of PERMA (positive emotions, engagement, relationships, meaning, accomplishment) and in the satisfaction with life scale. Using scale reliability analyses, we found that some of the dimensions in the PERMA model showed relatively low values both for Cronbach alpha and composite reliability. Several confirmatory structural equation models (single factorial, second order, and five factors) were calculated, as well as the usual adjustment measures, with the five-factor PERMA model presenting the best structure, although with a relatively low fit. The modification of the model by the weight of regressions between some of the items with larger covariances allowed a better adjustment: $\chi^2(73) = 264.960$, $p < 0.001$, $\chi^2/\mathrm{df} = 3.63$, CFI = 0.94, TLI = 0.91, IFI = 0.94; GFI = 0.92, RMSEA = 0.078, $p < 0.001$. Although the results revealed that the experiences of senior tourists when visiting the island of São Miguel contributed significantly to their well-being and the modified model presented superior adjustment quality, future studies are suggested to evaluate the quality of the PERMA model applied to tourism.

**Keywords:** well-being; positive emotions; senior tourism; PERMA; Azores

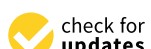

## 1. Introduction

Well-being in tourism experiences has been the target of recent attention in the academic community [1–4], with divergent results, the impact of vacations pointing either to an increase or decrease in the well-being and quality of life of the individual [5]. Nonetheless, recent studies have reported that senior tourists travel to enrich themselves personally, have new experiences, socialize, entertain and seek well-being [6–9], taking into account the existence of some concerns when planning a trip [10].

Research carried out on senior tourism in the Azores archipelago revealed that senior tourists who visited the Azores showed high levels of satisfaction with the trip [7,8]. According to Woo et al. [11], the satisfaction of tourists when making a trip affects the social,

cultural, family, and well-being domains. Such satisfaction may be due to tourist experiences increasing the feeling of involvement, pleasure, competence and personal growth, positive relationships, and health promotion [1], with evidence of a significant association between satisfaction with travel, satisfaction with life, and health perception [6,12].

Meanwhile, Han et al. [2] revealed a gap in research dedicated to well-being tourism, suggesting the need for more research regarding the impact of tourism on the psychological well-being of tourists, since there were similarities between psychological well-being between Eastern and Western cultures. In fact, there is a well-established relationship between tourism, travel, and health, in which the individual seeks relaxation and a greater sense of well-being [13]. Recent literature shows that senior tourists who go on frequent trips get a range of psychological benefits that contribute to eudaimonic (deep) happiness [14], where well-being can involve a wide range of human abilities and experiences [15].

## 2. Authentic Happiness, Psychological Well-Being, and Tourism

Positive psychology has been affirmed in the last two decades. In 2000, Seligman and Csikszentmihalyi [16] stated that the science of subjective positive experiences, positive individual traits, and positive institutions can promote quality of life and prevent diseases in the face of a sterile and meaningless life. Those authors also argued for the existence of human forces capable of softening mental illness (e.g., courage, optimism, faith, hope, honesty, perseverance).

Seligman [17] argued that authentic happiness can be analyzed through three distinct elements (positive emotion, commitment and meaning), allowing the individual to increase the amount of happiness in his life. Each of these three elements are predictors of life satisfaction [18] and considered the basis of well-being [19]. Therefore, well-being is considered a multidimensional construct, in which happiness and satisfaction with life cannot be universally indicative [20]; however, for Pawelski [21], the increase in happiness is not only for one individual but for all individuals. The results of the study by Paniagua et al. [22] revealed that tourists associate happiness with the dimensions of quality of life, satisfaction with life, and well-being. In fact, these areas seem to contribute to a better understanding of tourism for happiness [23], which Tien et al. [3] designated as the "economy of happiness".

Well-being is constituted through five measurable elements (Positive Emotion, Engagement, Relationships, Meaning and Accomplishment)—which configure the PERMA profiler model, in which each of these elements holds properties that contribute to the formation of well-being [17,24]. From another perspective, Diener et al. [25] argued that subjective well-being is defined by the affective and cognitive evaluation of the individual himself, in which positive relationships with events of personal fulfilment are involved.

Faced with two views on well-being, Goodman et al. [26] compared the PERMA model of well-being suggested by Seligman and the subjective well-being model suggested by Diener, verifying a latent and almost perfect correlation between the two proposed models. In this sense, Butler and Kern [27] suggested the non-acceptance of a single model that evaluates well-being, but rather different concepts that can contribute to the abstract construction of the concept of well-being and concentrate on the evaluation of positive emotions, commitment, relationship, meaning, and achievement.

Tourism has been recognized for offering health benefits, and there is evidence that the motivations to participate in this activity relate to ideas of well-being and health [13,14], with evidence that tourism promotes positive experiences [1,6], self-knowledge and well-being [6,9,28,29].

Leisure travel can have a positive impact on the subjective well-being of tourists [13], where well-being can influence the intention of tourists to revisit the site and positively suggest the visit (word of mouth) [4,30]. Thus, considering the existence of a strong evolution in the understanding of subjective well-being [31] and evidence of this in tourism, the present study aims to present a confirmatory analysis of the PERMA model in a sample

of Portuguese senior tourists who visited the island of São Miguel in the Autonomous Region of the Azores.

## 3. Methodology

### *3.1. Procedures*

The project "Senior Tourism: Welfare Routes and Local Experiences in an Island Ecosystem" (TURIVIVA+), developed on the island of São Miguel, in which the present study is included, was submitted to the evaluation of the ethics committee of the University of the Azores, having obtained a positive declaration with the reference 6/2022.

Subsequently, a request was sent to the Gorreana Tea Factory (a tourist attraction with a high tourist concentration) to authorize the application of the questionnaires to a sample of tourists. The researchers explained the study objectives to senior tourists (55 years or older) and guaranteed all protection measures in the face of SARS-CoV-2. Participation was voluntary in response to an instrument protocol with a set of questions of sociodemographic characterization, subjective well-being (PERMA Profiler), and life satisfaction (SWLS). A total of 1083 senior tourists of various nationalities participated. The inclusion criteria for the present study were: (i) having Portuguese nationality, and (ii) answering all the questions.

### *3.2. Instruments*

A questionnaire was elaborated regarding the sociodemographic characterization of the sample (e.g., variables of age, gender, marital status, level of education, physical limitations and others) and information related to their trip (e.g., type of hotel, with whom they traveled, a form of reservation, and type of accommodation).

### 3.2.1. PERMA Profiler

The PERMA model evaluates five dimensions of well-being (Positive Emotion, Engagement, Relationships, Meaning and Accomplishment). Butler and Kern [27] presented the development and validation of the PERMA Profiler, using a sample of 3500 participants to reduce the scale to 15 items with 10 response options in which (0) corresponds to never/terrible/absolutely nothing and (10) corresponds to always/excellent/completely. Each dimension consists of three items. The translation of PERMA Profiler for the Portuguese population was presented in 2015 by Alves et al. [32], revealing a valid psychometric measure for the investigation of Psychological Flowering.

### 3.2.2. Satisfaction with Life Scale (SWLS)

The scale of satisfaction with life was developed by Diener et al. [33] and adapted for the Portuguese population by Simões [34]. Through five items, the scale assesses overall satisfaction with life on a Likert scale with five points, from complete disagreement (1) to complete agreement (5). In the sample selected for the present study, the SWLS scale presented a Cronbach's alpha value considered as good ($\alpha = 0.81$).

### *3.3. Participants*

The eligible sample consisted of 434 participants with a mean age of 63.78 years (SD = 6.98), of which 51.8% were female. Table 1 shows other sociodemographic characteristics and travel information.

**Table 1.** Socio-demographic, economic and travel characteristics of a sample of 434 Portuguese senior tourist participants in the study, recruited at Gorreana Tea Factory, São Miguel Island, Azores.

| | Frequencies | |
|---|---|---|
| | *n* | *%* |
| **Marital status** | | |
| Married | 313 | 72.1 |
| Single | 30 | 6.9 |
| Divorced | 41 | 9.4 |
| Widower | 43 | 9.9 |
| De facto union | 3 | 0.7 |
| **Being retired** | | |
| Yes | 197 | 45.4 |
| No | 233 | 53.7 |
| **Education level** | | |
| Up to 4th class | 62 | 14.3 |
| 9th grade | 75 | 17.3 |
| Secondary education or equivalent | 84 | 19.4 |
| Technical or medium course | 26 | 6 |
| Bachelor | 38 | 8.8 |
| License degree | 98 | 22.6 |
| Master degree | 38 | 8.8 |
| PhD | 10 | 2.3 |
| **Economic income** | | |
| Live very well | 45 | 10.4 |
| Live comfortably | 244 | 56.2 |
| You can live | 134 | 31.2 |
| Living with difficulties | 4 | 0.9 |
| Living with many difficulties | 2 | 0.5 |
| **With whom you traveled** | | |
| In a group organized by a travel agency | 61 | 14.1 |
| With small group friends | 61 | 14.1 |
| With family | 170 | 28.6 |
| Alone | 14 | 3.2 |
| **How you booked the trip** | | |
| Travel agency | 125 | 28.8 |
| Internet | 230 | 53 |
| Through a family member/friend | 72 | 16.6 |
| **Type of accommodation** | | |
| Hotel | 251 | 57.8 |
| Local accommodation | 106 | 24.4 |
| Rural tourism | 5 | 1.2 |
| Friends/family house | 56 | 12.9 |
| Other type of accommodation | 2 | 0.5 |

Most participants lived with another person (*n* = 359, 82.7%) and had no physical limitations (*n* = 382, 88%). Asked if they had already visited the Azores, 47.9% reported having visited the Azores at least once before, and 77.4% of the participants did not use any tourist guide to visit the island of S. Miguel.

*3.4. Statistical Analyses*

We used IBM SPSS for Macintosh, version 28 (Armonk, New York, NY, USA) to calculate several statistical analyses: Kaiser-Meyer-Olkin (KMO) Measure of Sampling Adequacy and Bartlett's test of Sphericity; Cronbach's alpha for the scales, including the value for the entire scale, the values for the respective dimensions, and the item-total

correlations; and IBM SPSS AMOS for Windows, version 28 (Armonk) to evaluate the different structural equation models (SEM).

## 4. Results and Discussion

### *4.1. Reliability*

The PERMA model consisting of 15 items, distributed equally over five dimensions, presented a good sampling adequacy for factor analysis (KMO = 0.86). Bartlett's check ($\chi^2$(105) = 2,234,614; $p < 0.001$) revealed that the variables were sufficiently correlated. The item-total correlations were moderate to high, ranging from 0.56 to 0.81. The values of Cronbach's alpha were considered as unacceptable for the dimensions Engagement ($\alpha = 0.46$) and Relationship ($\alpha = 0.59$), low for the dimensions Accomplishment ($\alpha = 0.66$) and Positive Emotions ($\alpha = 0.77$) and moderate for the Meaning dimension ($\alpha = 0.82$). However, Marôco and Garcia-Marques [35] argued that the validity of this measure has been questioned, so the analysis of composite reliability is suggested.

Construct Reliability

To estimate whether the internal consistency of the items were consistent manifestations of the latent factor, the composite reliability was calculated ($\hat{FC}$), and an appropriate construct reliability was found for the Relationship ($\hat{FC} = 0.71$), Positive Emotion ($\hat{FC} = 0.75$) and Meaning ($\hat{FC} = 0.81$) dimensions. Although the dimensions Accomplishment ($\hat{FC} = 0.68$) and Engagement ($\hat{FC} = 0.51$) presented lower composite reliability values, Marôco [36] states that values below 0.70 are acceptable for exploratory studies. The confirmation of the factorial structure of the Portuguese version of PERMA Profiler also showed low values of Cronbach's alpha and composite reliability [32]. These differences may be due to the fact that sociocultural factors influence the language of an individual and their choice of words and hinder expression of their identity [37]. On the other hand, the translation of the items on the respective scale may also have contributed to the distortion of the meaning of the statements.

### *4.2. PERMA Profiler Model Validation with Structural Equation Models*

Significant relations between the structures in the models have been evaluated. The assumptions related to this analysis were checked before the SEM analysis was started. The sample size, multivariate normality, and multicollinearity assumptions required for this analysis have been tested. The skewness and kurtosis values of each variable were calculated for univariate normality, which was a prerequisite for meeting the assumption of multivariable normality. The usual model adjustment indexes were used in evaluating the model fit: $\chi^2$/df, CFI, GFI, TLI, NFI, IFI, RMSEA and SRMR [36].

The structure of the dimensions of the PERMA model showed adjustment quality indices considered as low—$\chi^2$(80) = 464.251, $p < 0.001$, $\chi^2$/df = 5.803, CFI = 0.83, TLI = 0.84, IFI = 0.83, GFI = 0.83, RMSEA = 0.12, $p < 0.001$—in which the regressions and variances of all items were significant ($p < 0.001$). In this sense, we chose to analyse the structure of a unifactorial model—$\chi^2$(90) = 649.859, $p < 0.001$, $\chi^2$/df = 7.221, CFI = 0.83, TLI = 0.84, IFI = 0.83, GFI = 0.83, RMSEA = 0.12, $p < 0.001$)—and a second-order model—$\chi^2$(85) = 574.026, $p < 0.001$, $\chi^2$/df = 6.753, CFI = 0.85, TLI = 0.81, IFI = 0.85, GFI = 0.82, RMSEA = 0.12, $p < 0.001$—with a weaker adjustment in these models than in the previous one. Thus, the initial model was respecified using the weight of regressions between some of the items, selected by the largest covariances represented in Figure 1.

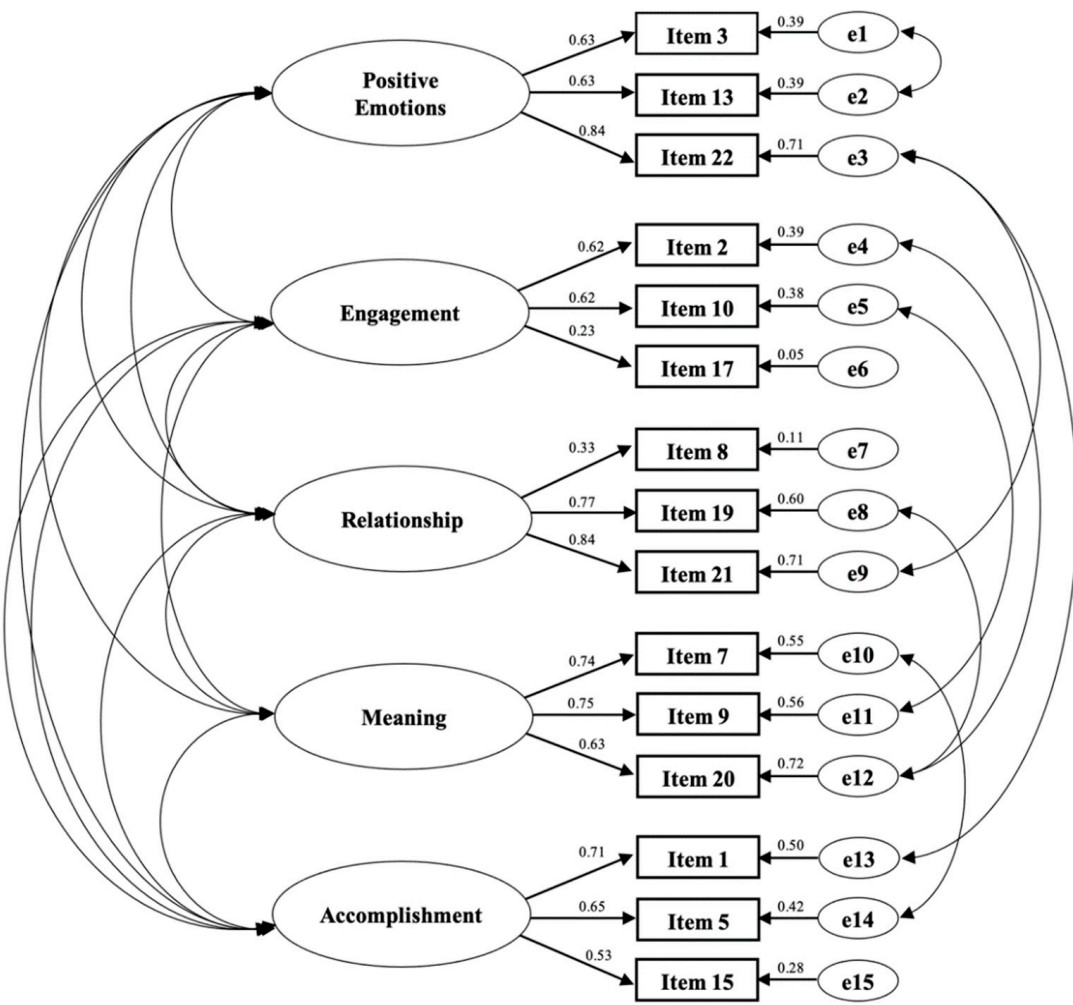

**Figure 1.** Confirmatory Factor Analysis of the PERMA model using a structural equation model with the following adjustment parameters: $\chi^2(73) = 264.960$, $p < 0.001$, $\chi^2/df = 3.63$, CFI = 0.94, TLI = 0.91, IFI = 0.94, GFI = 0.92, RMSEA = 0.078, $p < 0.001$.

After refining the model by the errors of the covariances between items 1–2, 3–9, 4–12, 3–13, 5–11, 8–12 and 10–14, it was possible to obtain a better adjustment: $\chi^2(73) = 264.960$, $p < 0.001$, $\chi^2/df = 3.63$, CFI = 0.94, TLI = 0.91, IFI = 0.94, GFI = 0.92, RMSEA = 0.078, $p < 0.001$.

The re-specification of the model allowed us to verify that the correlations between errors (residues) seemed to indicate that the corresponding items shared the same information. However, some authors (e.g., Bollen [38]; Gerbing and Anderson, [39]) maintain that one can move towards a model of hierarchical factors of higher order when: (i) there are substantial correlations between the factors; (ii) there are correlations between errors related to items that saturate in different factors; and (iii) when there is theoretical support. In fact, through Table 2, it was possible to confirm that all items were correlated ($p < 0.001$) in all dimensions. For Seligman [17,24], each element of PERMA can be defined and measured independently, and no isolated element can define well-being, arguing that all elements contribute to psychological well-being.

**Table 2.** Correlations Between Items and Dimensions of PERMA.

| | P | E | R | M | A | Item 3 | Item 13 | Item 22 | Item 2 | Item 10 | Item 17 | Item 8 | Item 19 | Item 21 | Item 7 | Item 9 | Item 20 | Item 1 | Item 5 | Item 15 |
|---|---|---|---|---|---|---|---|---|---|---|---|---|---|---|---|---|---|---|---|---|
| P | - | 0.53 *** | 0.55 *** | 0.69 *** | 0.60 *** | 0.83 *** | 0.86 *** | 0.78 *** | 0.42 *** | 0.61 *** | 0.29 *** | 0.31 *** | 0.56 *** | 0.63 *** | 0.57 *** | 0.57 *** | 0.62 *** | 0.46 *** | 0.49 *** | 0.47 *** |
| E | | - | 0.40 *** | 0.56 *** | 0.53 *** | 0.43 *** | 0.44 *** | 0.44 *** | 0.64 *** | 0.66 *** | 0.79 *** | 0.27 *** | 0.36 *** | 0.39 *** | 0.47 *** | 0.50 *** | 0.42 *** | 0.47 *** | 0.41 *** | 0.40 *** |
| R | | | - | 0.63 *** | 0.37 *** | 0.41 *** | 0.39 *** | 0.63 *** | 0.32 *** | 0.43 *** | 0.22 ** | 0.83 *** | 0.71 *** | 0.70 *** | 0.52 *** | 0.50 *** | 0.59 *** | 0.29 *** | 0.29 *** | 0.32 *** |
| M | | | | - | 0.60 *** | 0.52 *** | 0.56 *** | 0.70 *** | 0.48 *** | 0.69 *** | 0.29 *** | 0.38 *** | 0.63 *** | 0.66 *** | 0.85 *** | 0.84 *** | 0.82 *** | 0.48 *** | 0.47 *** | 0.51 *** |
| A | | | | | - | 0.48 *** | 0.55 *** | 0.48 *** | 0.57 *** | 0.52 *** | 0.28 *** | 0.19 *** | 0.40 *** | 0.45 *** | 0.57 *** | 0.48 *** | 0.46 *** | 0.83 *** | 0.81 *** | 0.63 *** |
| Item 3 | | | | | | - | 0.58 *** | 0.50 *** | 0.37 *** | 0.49 *** | 0.21 *** | 0.27 *** | 0.38 *** | 0.42 *** | 0.43 *** | 0.46 *** | 0.42 *** | 0.41 *** | 0.42 *** | 0.32 *** |
| Item 13 | | | | | | | - | 0.56 *** | 0.35 *** | 0.51 *** | 0.26 *** | 0.21 *** | 0.44 *** | 0.46 *** | 0.49 *** | 0.46 *** | 0.48 *** | 0.40 *** | 0.45 *** | 0.45 *** |
| Item 22 | | | | | | | | - | 0.35 *** | 0.56 *** | 0.25 *** | 0.32 *** | 0.63 *** | 0.79 *** | 0.55 *** | 0.55 *** | 0.72 *** | 0.35 *** | 0.39 *** | 0.45 *** |
| Item 2 | | | | | | | | | - | 0.41 *** | 0.21 *** | 0.23 *** | 0.26 *** | 0.34 *** | 0.45 *** | 0.40 *** | 0.34 *** | 0.53 *** | 0.46 *** | 0.40 *** |
| Item 10 | | | | | | | | | | - | 0.28 *** | 0.22 *** | 0.48 *** | 0.52 *** | 0.55 *** | 0.67 *** | 0.57 *** | 0.43 *** | 0.39 *** | 0.45 *** |
| Item 17 | | | | | | | | | | | - | 0.16 *** | 0.20 *** | 0.18 *** | 0.24 *** | 0.25 *** | 0.21 *** | 0.21 *** | 0.21 *** | 0.24 *** |
| Item 8 | | | | | | | | | | | | - | 0.31 *** | 0.31 *** | 0.35 *** | 0.34 *** | 0.26 *** | 0.19 *** | 0.16 *** | 0.12 *** |
| Item 19 | | | | | | | | | | | | | - | 0.66 *** | 0.49 *** | 0.46 *** | 0.70 *** | 0.30 *** | 0.28 *** | 0.41 *** |
| Item 21 | | | | | | | | | | | | | | - | 0.51 *** | 0.53 *** | 0.68 *** | 0.33 *** | 0.35 *** | 0.45 *** |
| Item 7 | | | | | | | | | | | | | | | - | 0.56 *** | 0.55 *** | 0.48 *** | 0.47 *** | 0.40 *** |
| Item 9 | | | | | | | | | | | | | | | | - | 0.60 *** | 0.37 *** | 0.38 *** | 0.44 *** |
| Item 20 | | | | | | | | | | | | | | | | | - | 0.35 *** | 0.31 *** | 0.48 *** |
| Item 1 | | | | | | | | | | | | | | | | | | - | 0.56 *** | 0.33 *** |
| Item 5 | | | | | | | | | | | | | | | | | | | - | 0.33 *** |
| Item 15 | | | | | | | | | | | | | | | | | | | | - |

Note: *** $p < 0.001$.

Tourism has been recognized as a healthy physical and mental activity [40] and vacations have been considered a way for individuals to move away from their daily routine, seeking to rest, relax and rejuvenate [23]. In fact, the results revealed significant and moderate correlations ($0.25 \leq |R_S| \leq 0.5$) between the scale of satisfaction with life and the dimensions of the PERMA model. These results may be due to the fact that well-being is a multifaceted construct that can be measured through a series of subjective and objective constructs [41], allowing an individual to subjectively believe that life is desirable, pleasant, and good [42].

In other validation studies, Cobo-Rendón et al. [42] found a good adjustment of the PERMA model in a sample of Chilean students, although Cronbach's alpha of the Engagement dimension was lower ($\alpha = 0.36$) when compared to our results ($\alpha = 0.46$). Watanabe et al. [43], in a validation study with a sample of Japanese workers, verified that both the unifactorial and the five-factor models, showed a poor adjustment, although the latter with the best fit. From the models analyzed (single-factor, higher-order, five-factor, and two-factor) in a version in the German language, the five-factor model showed the best adjustment [44]. The dissimilarities found in the validation of the PERMA model in different cultures may be because people's views of the world are increasingly diverse [45].

Despite the differences found in several studies on the validation of the PERMA model, Dillette et al. [5] argue that the PERMA model of subjective well-being applies to tourism. These authors support the hypothesis that the combination of hedonic and eudemonic measures can bring new value to tourism. Some authors argue that although senior tourists have idiosyncratic characteristics and different motivations when travelling, it is common for senior tourists to seek identity and well-being [7,8,10,12,15].

### 4.3. Measurement Invariance

One of the advantages of confirmatory factor analysis is the possibility of evaluating the invariance of the structure and parameters of a given instrument in several groups. In this sense, categorical variables were created to perform invariance tests for the variable sex (male—female) and the variable age class (empty nesters—young seniors—seniors). The invariance test was initiated with the adjustment of the confirmatory factor analysis model in each group, assuming that the model demonstrated adequate adjustment. The tests for invariance were performed considering the group classifications for equality of factor loadings of the items (metric invariance) and equality of item intercepts (scalar invariance).

### 4.3.1. Invariance Analysis in Gender

The models showed good adjustment: (i) gender configurated model, $\chi^2(146) = 420.632$, $p < 0.001$, CFI = 0.92, TLI = 0.88, RMSEA = 0.066, 90% CI [0.059, 0.073] and SRMR = 0.057; (ii) metric model, $\chi^2(156) = 435.224$, $p < 0.001$, CFI = 0.92, TLI = 0.89, RMSEA = 0.064, 90% CI [0.057, 0.072] and SRMR = 0.062; (iii) scalar model, $\chi^2(166) = 459.486$; $p < 0.001$, CFI = 0.91, TLI = 0.89; RMSEA = 0.064, 90% CI [0.057, 0.071] and SRMR = 0.063.

Taking into account the commonly used criterion $\Delta$CFI $\leq 0.01$ [46], the results indicated a similar adjustment as a result of the addition of equality restrictions concluding for the existence of metric ($\Delta\chi^2 = 14,592$, $\Delta$CFI = 0.00) and scalar ($\Delta\chi^2 = 24.262$, $\Delta$CFI = 0.01) invariance. The factorial structure and the intercepts of the indicators were similar for both male and female respondents. Gender invariance was also identified in the PERMA model adaptation for German culture [44].

### 4.3.2. Invariance Analysis in Age Group

Analyzing the model configurated for the age groups, a good adjustment was verified: $\chi^2(219) = 835.510$, $p < 0.001$, CFI = 0.94, TLI = 0.91, RMSEA = 0. 047, 90% CI [0.043, 0.055] and SRMR = 0.055. The adjustment was slightly different for the metric model: $\chi^2(239) = 568.381$, $p < 0.001$, CFI = 0.90, TLI = 0.87, RMSEA = 0.067, 90% CI [0.051, 0.073] and SRMR = 0.051. Considering the commonly used criterion, the results of the $\chi^2$ and

ΔCFI test indicated a significant decrease in the adjustment because of the addition of equality restrictions ($\Delta\chi^2$ = 4267.129, ΔCFI = 0.04), concluding non-metric invariance.

The adjustment of the scalar model, $\chi^2$(259) = 691.036; $p < 0.001$, CFI = 0. 93, TLI = 0.92; RMSEA = 0.044, 90% CI [0.040, 0.048] and SRMR = 0.081, suggested that the assumption of scalar invariance was not sustainable ($\Delta\chi^2$ = 122.655, ΔCFI = 0.03).

Therefore, the factorial structure and the intercepts of the indicators were not similar among the age classes for the present sample of senior tourists. Dong and Dumas [47], in a systematic review of the literature, found that age invariance may vary depending on how age groups are divided. In the present study, we considered the suggestion of a subdivision of senior tourists according to the study by Patterson and Balderas-Cejudo [48], considering empty nesters (55–64 years), young seniors (65–79 years), and seniors (80 years or more). On the other hand, multiple traditions within gerontology address the positive dimensions of the ageing process [49].

## 5. Conclusions

Senior tourism is a constantly growing activity, as tourists in this sector have greater financial capacity and more time to travel. However, the availability to travel depends on socio-demographic characteristics, and the socio-occupational or family situation (e.g., age, family, responsibilities, level of autonomy and physical and mental capacity) [50]. On the other hand, senior tourists present the search for self-knowledge and well-being as motivations to travel (see Medeiros et al., 2020, 2021; Silva, Medeiros, Moniz, et al., 2021; Silva, Medeiros, Vieira, et al., 2021).

After explaining the purpose of the study when approaching the senior tourists, we were confronted with remarkable life experiences (e.g., travelling to the island of São Miguel to mourn the loss of a spouse), intentions to buy a second home for holidays or simply smiles of great satisfaction. The PERMA model, although recent, has raised numerous investigations as to its applicability in subjective well-being; however, it is still a construct under considerable attention by the scientific community when applied to tourism.

From a theoretical point of view, this research revealed that the PERMA model can be applied to senior tourism. However, studies that have respecified the model through covariances are unknown, so the results should be interpreted with great caution.

Although the contributions of this study are significant, the collection of the sample took place during the resumption of tourism activity, still enduring the pandemic period of SARS-CoV-2 (COVID-19). Moreover, the methodological limitations inherent in the cross-cultural validation of a scale and the possible bias in the interpretation of the data presented are limitations to be considered for the improvement of future investigations involving the use of the PERMA model in tourism. Future studies are encouraged to understand whether tourism activities and memorable experiences can influence the well-being of tourists. On the other hand, there is a need to better understand the perception of senior tourists with regard to their physical and psychological needs during a leisure trip. Notwithstanding, stakeholders should focus synergies on (re)creating tourism products that promote the health and well-being of senior tourists.

**Author Contributions:** Investigation & Writing—original draft, J.M.; Writing—review & editing, T.M., O.S., L.T., L.S. and J.A.F. All authors have read and agreed to the published version of the manuscript.

**Funding:** The development of this study was financed under the research project "Senior Tourism: Well-being Routes and Local Experiences in an Island Ecosystem" (TURIVIVA+), by the program PO2020 Azores and Regional Government of the Azores, with the reference ACORES-01-0145-FEDER-000115.

**Institutional Review Board Statement:** Not applicable.

**Informed Consent Statement:** Informed consent was obtained from all subjects involved in this study.

**Data Availability Statement:** Not applicable.

**Conflicts of Interest:** The authors declare no conflict of interest.

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
