# Peer review of "PERMA Model of Well-Being Applied to Portuguese Senior Tourists: A Confirmatory Factor Analysis"

_sustainability, doi:10.3390/su14137538_

Round 1

Reviewer 1 Report

Great article. It would really add to the research if you collected discourse about experience as well to bring the quantitative data to life. It would be beneficial to expand the conclusions a bit more. 

Author Response

Dear reviewer, thank you for your comments.
Indeed, your observations are relevant. During sample collection, many tourists mentioned choosing São Miguel Island for providing them with well-being, however, the present study aims to present the psychometric properties of the PERMA-Profiler in a sample of senior tourists. 

I can affirm, that at least two tourists mentioned that they were taking this trip to help them in the grieving process for the loss of their spouse.

Indeed, future studies should effectively assess subjective well-being before and after the trip, but here the linking of synergies between stakeholders is missing.

The conclusion has been improved.

Regards

Reviewer 2 Report

the article is of some interest. But there are some comments

1. it is necessary to strengthen the relevance of the chosen topic.

2. Designate the category "elderly tourists".

3. It is not clear how subjective assessment can correlate with the feelings of tourists themselves.

4. The concept of "well-being". What the authors mean.

5. It is necessary to give a detailed analysis of not just the respondents' answers. It is necessary to make a "social" portrait and draw conclusions.

 6. there is a discrepancy in numbers. Abstract - 434 , p . 3 -433 participants/

7.   it is not clear why^ SWLS were applied to 1083 senior tourists ( 55 years) of various nationalities. To  evaluate the PERMA model for senior tourism in Portugal, a total of 434 senior tourists of Portuguese nationality were extracted from the sample

8. why is it being considered exactly the island of São Miguel&

9. what is the novelty of this study? what did the authors want to show readers? Where can the results of this study be used?

Author Response

Dear reviewer, thank you for your careful reading and valuable comments.
All suggestions have been taken into consideration, as you can see in the revision.

Regards

Reviewer 3 Report

The article is well written and the analyses are well done. However, I think that instead of applying SEM analysis with AMOS it would be better to apply GSEM analysis (Generalized Structural Equation Modeling) with Stata. This would allow us to analyze the influence of socio-demographic, economic and travel characteristics on factors. 

Author Response

Dear reviewer, thank you for your comments.

Indeed, your indications are extremely relevant, however, the present study only intends to present the confirmatory analysis of the PERMA-Profile in a sample of senior tourists. 

In a future article we intend to analyze these variables together with the scale of memorable experiences, also applied to the same sample.
The present suggestions have given us an orientation for the next article. In that, we will use the GSEM analysis (Generalized Structural Equation Modeling).

Regards, 

Round 2

Reviewer 2 Report

dear authors. It can be seen that you worked according to the recommendations